# Refined Grain Enhancing Lithium-Ion Diffusion of LiFePO$_4$ via Air Oxidation

**Xinjie Shen** [1,2], **Zijun Qin** [1,2], **Peipei He** [1,2], **Xugang Ren** [1,2], **Yunjiao Li** [1,2,*], **Feixiang Wu** [1,2], **Yi Cheng** [1,2] and **Zhenjiang He** [1,2,*]

1   School of Metallurgy and Environment, Central South University, Changsha 410083, China; 203512139@csu.edu.cn (X.S.); 223512126@csu.edu.cn (Z.Q.); 223512103@csu.edu.cn (P.H.); 203501061@csu.edu.cn (X.R.); feixiang.wu@csu.edu.cn (F.W.); yi_cheng@csu.edu.cn (Y.C.)

2   National Engineering Research Center of Low-Carbon Nonferrous Metallurgy, Central South University, Changsha 410083, China

*   Correspondence: yunjiao_li@csu.edu.cn (Y.L.); hzjcsu@csu.edu.cn (Z.H.); Tel.: +86-151-111-886-80 (Y.L.)

**Abstract:** LiFePO$_4$ is a type of cathode material with good safety and long service life. However, the problems of the low Li ion diffusion rate and low electron conductivity limit the application of LiFePO$_4$ in the field of electric vehicles. In this paper, FePO$_4$ with different grain sizes was prepared via the air oxidation precipitation method and then sintered to prepare LiFePO$_4$. The refined grain can shorten the diffusion distance of Li$^+$, accelerate the diffusion of Li$^+$, and improve the diffusion coefficient of Li$^+$. The results show that LiFePO$_4$ with a smaller grain size has better electrochemical performance. The discharge capacity of the first cycle is 151.3 mAh g$^{-1}$ at 1 C, and the capacity retention rate is 95.04% after 230 cycles. Its rate performance is more outstanding, not only at 0.2 C, where the discharge capacity is as high as 155 mAh g$^{-1}$, but also at 10 C, the capacity fade is less, and it can still reach 131 mAh g$^{-1}$. The air oxidation precipitation method reduces the production cost, shortens the production process, and prepares FePO$_4$ with small grains, which provides a reference for further improving the properties of precursors and LiFePO$_4$.

**Keywords:** iron phosphate; lithium iron phosphate; refined grain





## 1. Introduction

With energy shortage and environmental issues receiving much attention, it is becoming increasingly urgent to develop and utilize green and environmentally friendly new energy sources. Lithium-ion batteries are one of the most popular electrochemical energy storage systems, characterized by safety, high efficiency, and being green [1,2]. The cathode material component is the most critical link; it is not only the material involved in the electrode reaction, but also directly affects the cycle life, energy density, and power density of Li-ion battery cells [3]. Since LiFePO$_4$ with olivine structures was first reported by the Goodenough group in 1997, LiFePO$_4$ has been considered as one of the most promising lithium-ion cathode material because of its good cycle performance, long service life, and high safety performance [4,5]. In recent years, the development momentum of new energy vehicles in the whole society is flourishing. The power battery and energy storage battery industry has ushered in a broader space for development, and the market demand for lithium-ion battery cathode materials is strong [6]. As a result of the decline in the ternary materials subsidy policy, as well as the promotion and application of new technologies such as Contemporary Amperex Technology Co., Limited (CATL, Ningde, China) CTP technology and Build Your Dreams (BYD, Shenzhen, China) blade battery technology, the performance of lithium iron phosphate battery has been greatly improved, and the cost advantage has been further highlighted, which has been more widely used in power batteries [7]. In the crystal structure of LiFePO$_4$, the FeO$_6$ octahedron, LiO$_6$ octahedron, and PO$_4$ tetrahedron alternate in sequence to form a lamellar structure in the direction along

the a-axis [8,9]. From the bc plane, each $FeO_6$ octahedron is connected to the surrounding four $FeO_6$ octahedron by a common vertex, forming a jagged planar layer [10]. Along the c-axis direction in a chain pattern, one $PO_4$ tetrahedron and one $FeO_6$ octahedron and two $LiO_6$ octahedrons share the same edge, thus forming a three-dimensional spatial meshwork structure [11]. The $PO_4$ tetrahedron is connected to the parallel $FeO_6$ octahedron, forming only a narrow one-dimensional $Li^+$ diffusion channel, which limits the insertion and extraction of $Li^+$ during charging and discharging [12]. Due to the adjacent $FeO_6$ octahedrons are connected to each other by common vertexes, there is no metric to form a continuous mesh, resulting in poor electronic conductivity of $LiFePO_4$ [13]. The inherent drawbacks of $LiFePO_4$ hinder its application in the field of passenger cars [14–16]. These two shortcomings lead to the serious capacity attenuation and even almost no capacity of $LiFePO_4$ when discharging at high rates. At the same time, the precursor $FePO_4$ is also facing the problem of increasing production costs, and there is an urgent need to develop a low-cost process for the preparation of $FePO_4$.

In order to address the disadvantages of $LiFePO_4$ and reduce the production cost of the precursor $FePO_4$, various strategies have been proposed [17]. Among these, surface coating and the control of the particle size and morphology are considered to be effective measures to solve the inherent defects of $LiFePO_4$ [18,19]. Carbon coating can inhibit the growth of $LiFePO_4$ particles and increase the electronic conductivity of $LiFePO_4$, thus effectively improving the rate performance of $LiFePO_4$ [20–22]. Liu et al. synthesized $LiFePO_4$/carbon/graphene composites via the solvothermal method, which significantly improved the cycle performance and rate performance [23]. They attribute this to the excellent conductivity of graphene and the excellent conductive network formed by co-modification of graphene with carbon coating. However, there is also some literature suggesting that carbon modification does improve the overall conductivity of $LiFePO_4$ electrodes, but the determining step in the rate capability has not yet been determined [24]. Surface coating of the carbon conductive layer only changes the conductivity between different $LiFePO_4$ particles, and does not improve the structure of $LiFePO_4$ at the molecular level [25,26]. Therefore, it is difficult to obtain the $LiFePO_4$ material with a high-rate performance only through carbon coating technology. Some studies have shown that spherical particles have greater fluidity, which is beneficial to obtain more contact points [27] and excellent electrochemical performance [28–30]. Cheng et al. prepared nano-sized $LiFePO_4$ cathode materials using the two-step solid phase method, which effectively improved the electrochemical performance [31]. The electrochemical performance of $LiFePO_4$ was significantly improved by reducing particle size [32]. This is mainly because the diffusion path length of $Li^+$ in the electrode material is reduced. In addition, the small grain size leads to a reduction in the overall charge transfer resistance of the electrode and an increase in the diffusion area of the $Li^+$ [33–35]. In some literatures, the reduction in the particle size of $LiFePO_4$ is considered as the decisive factor to increase the discharge capacity of the material [36]. Wang et al. prepared olivine type $LiFePO_4$ nanoplates via the glycol-based solvothermal method, which exhibited good specific capacity at 5 C and 10 C [37]. Reducing the production cost of the phosphorus source and iron source is a common means to reduce the production cost of $FePO_4$. Ma et al. prepared battery-grade $FePO_4$ by using waste phosphorus and iron slag as raw materials to provide sources of phosphorus and iron [38]. This method can effectively utilize the waste residue of $FePO_4$ and has certain commercial value.

However, industries usually use ferrous sulphate as a raw material to prepare ferric phosphate, but the existing process must consume large amounts of oxidants such as hydrogen peroxide and sodium hypochlorite, which increases the production costs to some extent. Moreover, few studies have combined reducing the production cost of $FePO_4$ with improving the electrochemical performance of $LiFePO_4$. In this paper, $FePO_4$ with different grain sizes was prepared via the air oxidation precipitation method. This method not only saves on the cost of purchasing oxidants, but also reduces the cost of subsequent waste liquid disposal. Moreover, the purity of the product is high, and the ratio of iron

to phosphorus is close to 1. In addition, the effect of the grain size on $LiFePO_4$ was also investigated. The $LiFePO_4$ synthesized from small grain $FePO_4$ has a better rate performance. This method provides a new idea for reducing the production cost of $FePO_4$ and improving the performance of precursor $FePO_4$ and $LiFePO_4$.

## 2. Materials and Methods

### 2.1. Raw Materials

All chemicals and materials used in this study are analytical-grade reagents (A.R.) (Sinopharm Chemical Reagent Shanghai Co., Ltd., Shanghai, China). They are ferrous sulfate heptahydrate ($FeSO_4 \cdot 7H_2O$), phosphoric acid ($H_3PO_4$), ammonia ($NH_3 \cdot H_2O$), lithium hydroxide monohydrate ($LiOH \cdot H_2O$), and glucose monohydrate ($C_6H_{12}O_6 \cdot H_2O$). Among them, $FeSO_4 \cdot 7H_2O$ and $H_3PO_4$ provide iron sources and phosphorus source respectively.

### 2.2. Synthesis of FePO₄ and LiFePO₄/C

Synthesis of precursor $FePO_4$: a certain mass of $FeSO_4 \cdot 7H_2O$ (Macklin, Shanghai, China) and $H_3PO_4$ (wt 85%) were weighed, and the molar ratio of $FeSO_4 \cdot 7H_2O$ to $H_3PO_4$ was 1:1. $FeSO_4$ solution was prepared by adding deionized water and oxidized by air for 30 min under strong stirring. Then, adding phosphoric acid to the solution at 323 K and 343 K respectively, and dropping $NH_3 \cdot H_2O$ to control the pH of the solution to 2. After a period of reaction, a white precipitate was obtained. The product was calcined at 823 K for 4 h to obtain crystalline $FePO_4$. The $FePO_4$ synthesized at 323 K is marked as FP−a and the $FePO_4$ synthesized at 343 K is marked as FP−b.

Synthesis of $LiFePO_4$: The self-made precursor, $LiOH \cdot H_2O$, $C_6H_{12}O_6 \cdot H_2O$, were poured into a ball mill and uniformly mixed, with a molar ratio of 1.05:1 for the precursor and $LiOH \cdot H_2O$, and a mass of 20% of the precursor mass for $C_6H_{12}O_6 \cdot H_2O$. FP−a and FP−b were evenly mixed with $LiOH \cdot H_2O$ and $C_6H_{12}O_6 \cdot H_2O$, respectively, with a molar ratio of 1.05:1 for the precursor and $LiOH \cdot H_2O$. The mass of $C_6H_{12}O_6 \cdot H_2O$ is 20% of the mass of the precursor. Finally, the mixed powder was calcined in nitrogen atmosphere at 953 K for 10 h to obtain $LiFePO_4$ cathode material. The $LiFePO_4$ prepared by FP−a was marked as LFP−a, and the $LiFePO_4$ prepared by FP−b was marked as LFP−b.

### 2.3. Characterization of FePO₄ and LiFePO₄/C

The element content of the synthesized material was determined using inductively coupled plasma-emission spectrometry (ICP) (Thermo Fisher Scientific Inc., Waltham, MA, USA). The morphology and coating of the material were characterized by using a scanning electron microscope (SEM) (Japan Electron Optics Laboratory, Tokyo, Japan) and transmission electron microscope (TEM) (FEI Company, Hillsboro, OR, USA). The crystal structure and phase composition were studied via X-Ray diffraction (XRD) (PANalytical B.V., Almelo, The Netherlands).

### 2.4. Electrochemical Measurement

The active substance (wt 80%), acetylene black (wt 10%), and polyvinylidene fluoride (PVDF, wt 10%) were evenly mixed with N-methyl-2-pyrrolidone. The mixed slurry is evenly coated with aluminum foil and dried for 6 h in a 363 K vacuum drying oven. The CR2016 coin half-cells used in electrochemical testing were assembled in a glove box filled with argon gas. The charge-discharge tests of the cells were carried out in the LAND test system (CT2001A) (Wuhan LAND Electronic Co.,Ltd, Wuhan, China) (version CT2001A) in the voltage range of 2.5–4.2 V at an ambient temperature of 298 K, and in this study, 1 C is equivalent to 170 mAh $g^{-1}$. The cyclic voltammetry (CV) curves were obtained under the conditions of 2.5–4.2 V and 0.1–0.5 mV $s^{-1}$. The electrochemical impedance spectroscopy (EIS) (Princeton Instruments, Trenton, NJ, USA) was carried out in the frequency range of 0.01–100 kHz by using the Princeton Electrochemical Work Station (Princeton Instruments, Trenton, NJ, USA).

## 3. Results

### 3.1. Characterization of Precursor FePO₄

The XRD patterns of the $FePO_4$ precursor obtained at different synthesis temperatures are shown in Figure 1a. The "//" at the number "22" in the coordinate axis represents a break point, indicating that the numbers are not continuous. It can be seen that the characteristic peaks formed by sample FP−a and FP−b in the range of 10–80° are basically consistent with the standard $FePO_4$ (PDF#77-0094). The synthesized $FePO_4$ corresponds to the trigonal system and belongs to the space group P3121. The diffraction peaks of the two samples are sharp, and there are no obvious diffraction peaks of impurity phases. This indicates the good crystallinity of the samples and the high purity of the phases. As can be seen from the partial magnified XRD pattern (Figure 1b), the peaks of FP–b are slightly shifted to the right, and the peaks are broadened. The particle size can be calculated according to Scherrer equation, which is given in the following Equation (1):

$$D = \frac{K\gamma}{B\cos\theta} \tag{1}$$

where K is the Scherrer constant (K = 0.89); D is the average thickness of the grain perpendicular to the crystal plane, and the unit is nanometers; B is the full width at half maximum (FWHM) of the measured sample diffraction peak, which needs to be converted into radians (rad) in the process of calculation; θ is the Bragg diffraction angle, and the unit is degree; γ is the X-Ray wavelength (γ = 0.154056 nm). The average grain sizes of FP−a and FP−b were calculated to be 46.9 and 40.3 nm, respectively. The smaller grain size of FP–b may be due to the increased thermal motion of ions as the temperature rises, increasing the number of possible combinations of different ions and facilitating the nucleation process. Then the contents of P and Fe in the two samples were measured using ICP (Thermo Fisher Scientific Inc., Waltham, MA, USA). As shown in Table 1, the Fe/P ratios of the both samples are close to 1.0, which further indicates that the product is purity without other impurities.

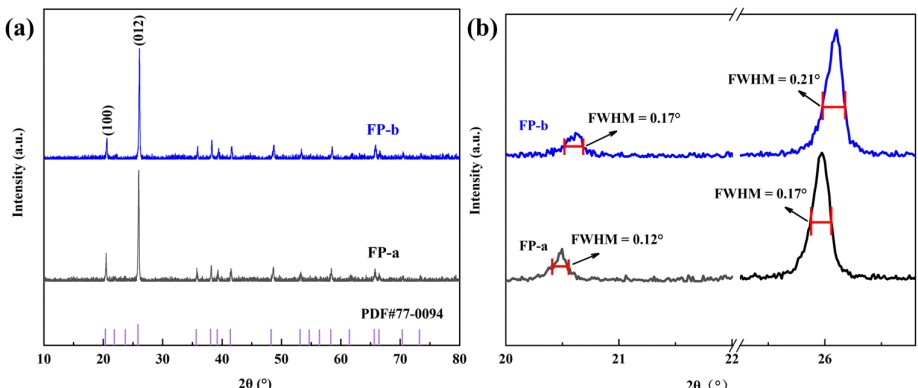

**Figure 1.** (**a**) XRD patterns of FP−a and FP−b; (**b**) Partial magnification of XRD for FP−a and FP−b.

**Table 1.** ICP test results of two samples.

| Sample | Fe (ppm) | P (ppm) | Fe/P (Molar Ratio) |
|--------|----------|---------|--------------------|
| FP-a | 72.682 | 40.375 | 0.9984 |
| FP-b | 75.535 | 41.079 | 1.0198 |

Then, the micro-morphologies of the two samples were characterized. As shown in Figure 2a–f, it can be seen that the particle size of FP−a is about 1 um, and that of FP−b is around 900 nm. The primary particle size of FP−a was 100−200 nm, and that of Fb−b was 300−400 nm, both of which showed regular shape of thin slices. This is probably because the primary particles of both FP−a and FP−b are made up of multiple grains bonded together. FP−b, however, has smaller grains, resulting in a larger number of grains

bonded and therefore a larger primary particle size. In addition, due to the small primary particle size of FP−a, multiple primary particles are heavily agglomerated in FP−a. In contrast, FP-b adsorbs some fine particles on the surface of the primary particles and the agglomeration is lighter.

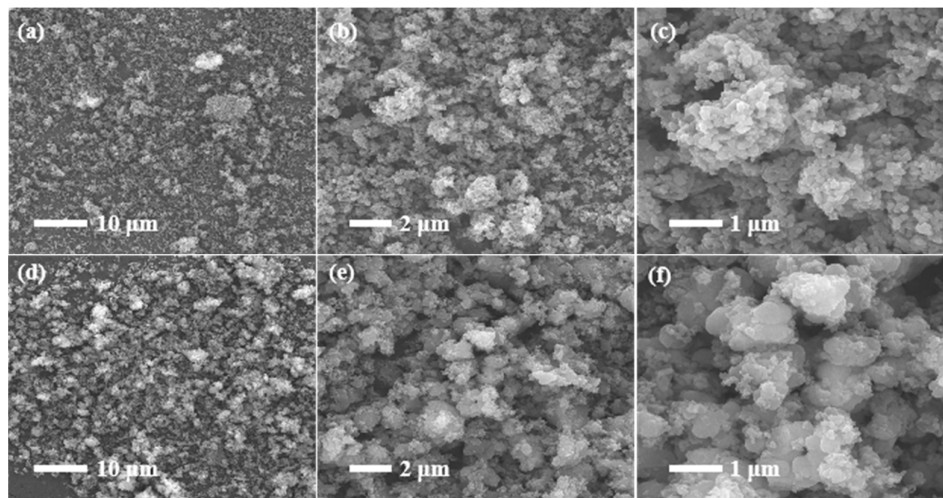

**Figure 2.** SEM images of the FP−a (**a**–**c**) and FP−b (**d**–**f**).

*3.2. Characterization of LiFePO$_4$/C*

Then, the crystal structure of the sintered products of the two samples was detected as shown in Figure 3a. The diffraction peaks of the two LiFePO$_4$ samples are in good agreement with PDF#83-2093, indicating that the crystal structure of the product belongs to the Pnma space group and is olivine structured LiFePO$_4$. Pnma is an orthogonal crystal system with a centrosymmetric space group. In the Pnma notation, p denotes a simple lattice, n denotes an n-slip surface, m denotes a mirror surface and a denotes an a-slip surface. There are no obvious impurity diffraction peaks in both samples, but the diffraction peak intensity of LFP−b is slightly weaker. It can be seen from the locally amplified XRD pattern (Figure 3b) that compared with LFP−a, the FWHM of LFP−b is increased. And the FWHM is inversely proportional to the grain size. According to the Scherrer equation, the grain sizes of LFP−a and LFP−b are 111.1 and 92.7 nm, respectively.

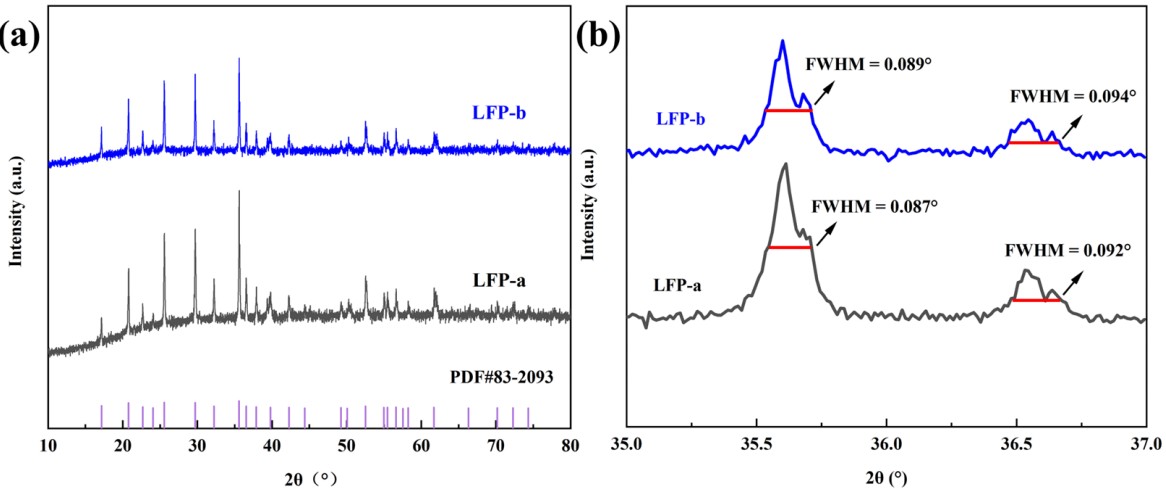

**Figure 3.** (**a**) XRD patterns of LFP−a and LFP−b; (**b**) Partial magnification of XRD for LFP−a and LFP−b.

The micro-morphologies of LFP−a and LFP−b are shown in Figure 4a−f. LFP−a has a particle size in the range of 600−700 nm, while LFP−b has a smaller particle size, ranging from 300 to 400 nm. This may be because the particle morphology and particle size of the FP largely determine the particle morphology and size of the LFP. FP−b has smaller particles and therefore the synthesized LFP−b particles are smaller. Moreover, LFP−b exhibits better particle size uniformity compared to LFP−a. The particle morphology of LFP−a is mainly block, while that of LFP−b is quasi-spherical. Many fine particles are stuck to the surface of the primary particles of the two samples.

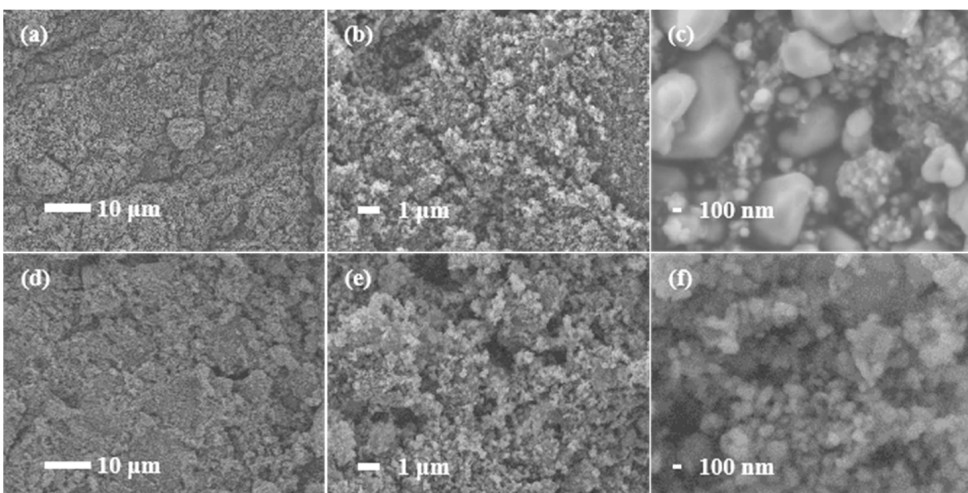

**Figure 4.** SEM images of the LFP−a (**a–c**) and LFP−b (**d–f**).

In order to further study the micro-surface/interface structure, two samples were tested using TEM. Figure 5a−d shows TEM images of two samples and corresponding high-resolution TEM (HR−TEM) images of high magnification. It can be seen that carbon coating exists on the surface of both samples and is coated on the periphery of the particles. In addition, the two samples show clear lattice fringes in the corresponding HR−TEM images, indicating that they have a good layered structure. At the same time, it can be seen from the fast Fourier transform (FFT) diagram that the two samples have good crystallinity. In LFP−a, the spacing of lattice fringes in the selected region is 3.811 Å, corresponding to the (011) crystal plane of $LiFePO_4$. However, in LFP−b, (011) crystal plane is the same, and the spacing of lattice fringes decreases to 3.751 Å. This also indicates that the grain size of LFP−b decreases somewhat.

In order to evaluate the electrochemical performance of the samples, the assembled half-cells were tested. Figure 6a,b shows the discharge curves of LFP−a and LFP−b at different current densities (0.5, 1, 2, 5 and 10 C). The arrows represent the gradual increase in current density from top to bottom. It can be seen that the discharge specific capacity of the two samples decreases with the increasing of current density. At the same current density, LFP−b has longer discharge platform and higher discharge specific capacity, which is due to grain refinement. Figure 6c,d shows the differential capacity in the voltage range of 3.20–3.62 V of LFP−a and LFP−b. It can be seen from the figure that there are two polarization peaks in both samples. The oxidation potential and reduction potential corresponding to these two polarization peaks provide the voltage value of the transition between $LiFePO_4$ and $FePO_4$ phases. The voltage difference of LFP−b under different cycles is smaller than that of LFP−a, indicating that the electrochemical polarization of LFP−b is smaller. The cycling curves of samples at the voltage range from 2.5 to 4.2 V, and the rate of 1 C is shown in Figure 6e. The discharge specific capacities of LFP−a and LFP−b in the first cycle at 1 C were 132.8 and 151.1 mAh $g^{-1}$, respectively. Moreover, the capacity retention rates after 230 cycles were 94.73% and 95.04%, respectively. In addition, the rate performances of the samples under different current densities were compared. As shown in Figure 6f,

the discharge specific capacity of LFP−b is higher than that of LFP−a at each discharge current density, and the difference of specific capacity increases at a high discharge rate. The discharge specific capacities of LFP−a and LFP−b at 10 C are 91 and 130 mAh g$^{-1}$, respectively, and the capacity retention rates are 70.8% and 83.5%, respectively. When the discharge current density is restored to 0.2 C, the capacity attenuation rates are 7.78% and 5.13%, respectively. The enhancement of the rate performance of LFP−b may be due to the shortening of the migration path of Li$^+$ after grain refinement. This is beneficial for reducing the impedance of Li$^+$ migration and promote the insertion/removal of Li$^+$ under high current.

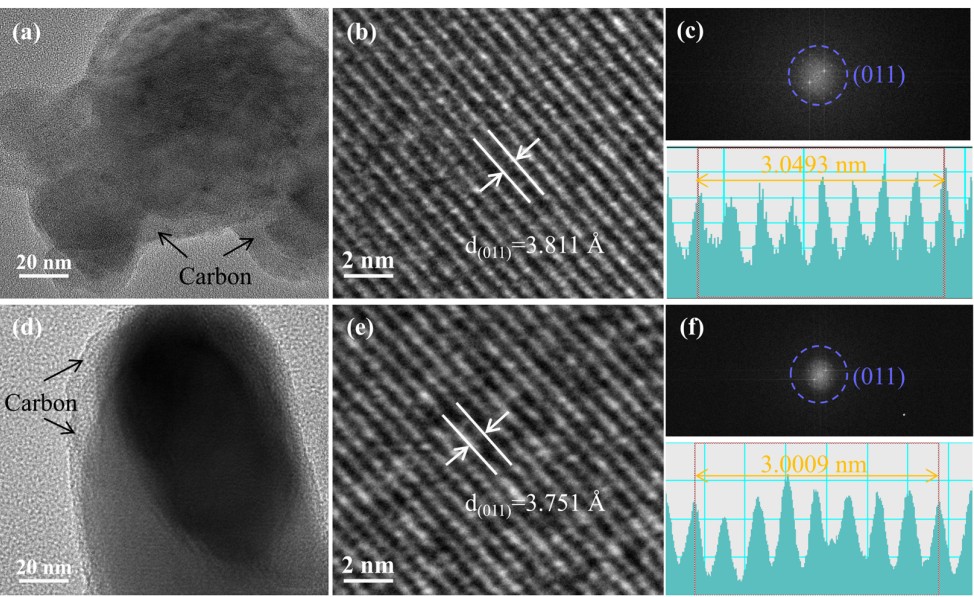

**Figure 5.** TEM images of the LFP−a (**a**) and LFP−b (**d**); (**b**,**e**) is the enlarged HR−TEM image of a certain area of two samples; (**c**,**f**) is the diffraction pattern obtained after the Fourier transform.

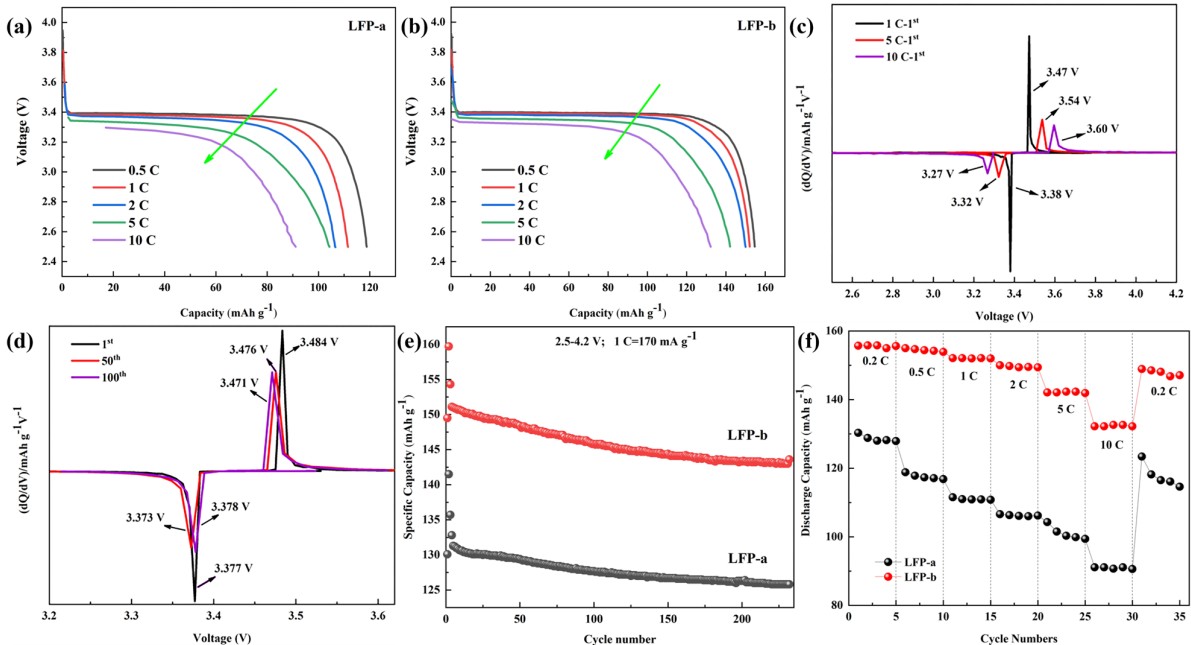

**Figure 6.** (**a**,**b**) The discharge curves of LFP−a and LFP−b at different current densities; (**c**,**d**) Differential curves at different cycles in the voltage range of 3.20–3.62 V of LFP−a and LFP−b; (**e**) cycle performance at 1 C/1 C; (**f**) rate performance.

The dQ/dV represents the voltage fluctuation of the material in the unit capacity range. After activation, the dQ/dV of the two samples charged/discharged in different cycles at different current density were obtained, as shown in Figure 7a,d. It can be clearly seen that both samples have a pair of oxidation and reduction peaks near 3.4 V, corresponding to the LiFePO$_4$/FePO$_4$ phase transition. The dQ/dV peak of the charge curve of the two samples moves towards the high voltage region with the increase of the current density. At the same current density, the difference between the oxidation peak and the reduction peak decreases with the increase in the number of cycles. This indicates that the loss of active lithium increases with the increase of charge-discharge times. As shown in Figure 7a,b, the voltage difference of LFP−b is smaller than that of LFP−a at the same multiplier in the first cycle. This indicates that the electrochemical polarization is reduced effectively by grain refinement and the ionic conductivity is better. This is because the reduction in particle size facilitates the diffusion of Li$^+$ as well as the transport of electrons. It can be seen from Figure 7c,d that with the increase in the number of cycles, the loss of active lithium increases, but the potential difference between the oxidation and reduction peaks of LFP−b is still smaller than that of LFP−a. The loss of active lithium may be due to the growth of the anode solid electrolyte interface (SEI) membrane caused by electrolyte decomposition and lithium depletion as the cycle continues. The smaller voltage difference for LFP−b is probably due to the fact that the small grain size makes the substance more structurally stable and the impedance increases slowly during cycling.

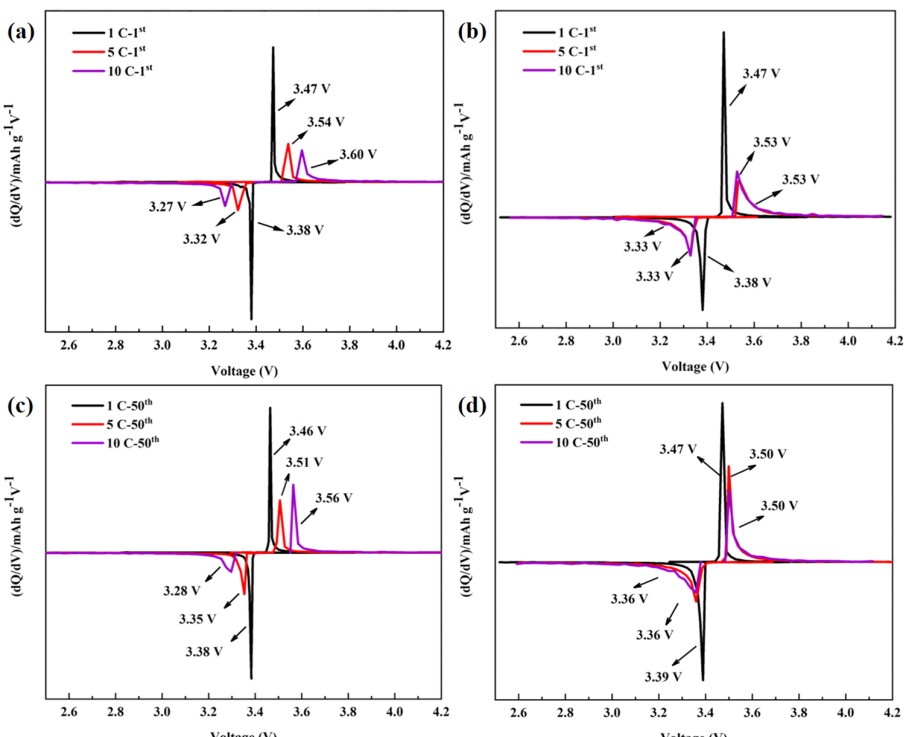

**Figure 7.** Differential curves of different periods of LFP−a and LFP−b at different rates in the voltage range of 2.5–4.2 V: (**a,b**) first cycle; (**c,d**) 50th cycle.

Figure 8a,b shows the CV curve of the sample at different scanning rates to study the electrochemical behavior of Li$^+$ in the electrode. Both samples showed a pair of relatively symmetrical peaks corresponding to the oxidation and reduction process of Fe$^{2+}$/Fe$^{3+}$. The oxidation and reduction peak voltage difference of LFP−a is 0.17 V, while that of LFP−b is 0.12 V, which indicates that LFP−b is more reversible and the structure and properties of the materials are more stable. Furthermore, compared with LFP−a, LFP−b has a sharper peak shape and a larger peak area, indicating that LFP−b has better Li$^+$ diffusion kinetics and higher material capacity. In order to further understand the kinetic process of the

electrode/interface, two samples were analyzed via EIS (Princeton Instruments, Trenton, NJ, USA). Figure 8c,d shows the Nyquist plots of the two samples before cycling and $Z'-\omega^{-1/2}$ plots of the low-frequency region. $R_s$ represents electrolyte impedance and $R_{ct}$ represents interface charge transfer impedance. It can be seen from Table 2 that the resistance of the sample is mainly from the charge transfer resistance $R_{ct}$. The $R_{ct}$ value of LFP−a was 70.8 Ω, while that of LFP−b was 56.7 Ω, which is due to the shortening of the migration path due to the grain refinement of LFP−b. In order to better show the effect of grain refinement, the diffusion coefficient of $Li^+$ was calculated according to Equation (2):

$$D(Li^+) = \frac{R^2T^2}{2An^4F^4\sigma^2C^2} \tag{2}$$

$$Z' = R_s + R_{ct} + \sigma\,\omega^{-1/2} \tag{3}$$

where R is the gas constant; T is the absolute temperature; A is the area of the electrode plate; n is the number of electrons gained or lost in the reaction; F is the Faraday constant; C is the concentration of lithium ion in the electrode; $Z'$ represents the real part of the impedance; σ is the Warburg impedance factor, that is, the slope in Equation (3); and ω represents the frequency of the impedance. The slope of the two lines fitted in Figure 7d can be used to calculate $D(Li^+)$. According to the calculation, before the cycle, the $D(Li^+)$ of LFP−a is $2.98 \times 10^{-14}$ and the $D(Li^+)$ of LFP−b is $3.40 \times 10^{-14}$, which shows that grain refinement shortens the migration distance of $Li^+$ and promotes the diffusion of $Li^+$.

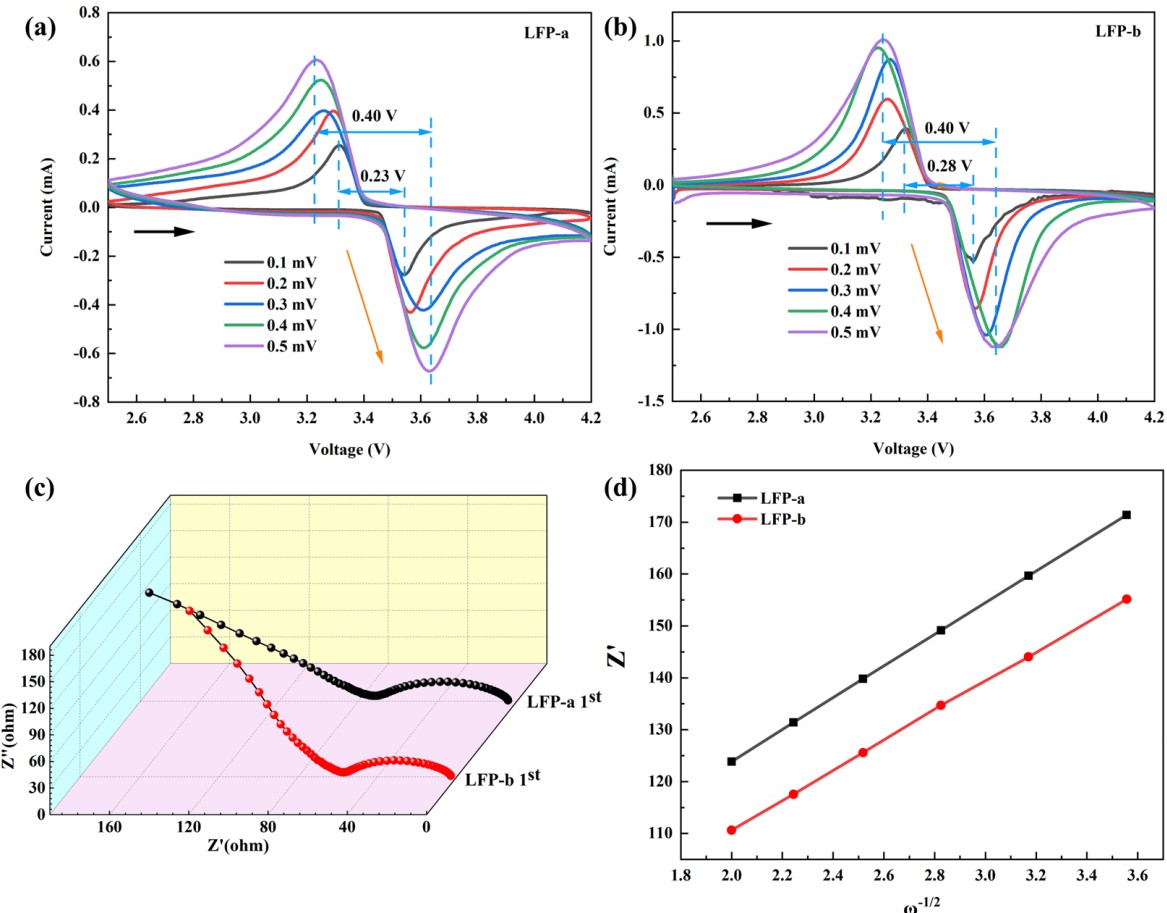

**Figure 8.** (**a**,**b**) CVs of LFP−a and LFP−b at different scan rate; (**c**) Nyquist plots of LFP−a and LFP−b before cycling. (**d**) $Z'-\omega^{-1/2}$ plots of the low-frequency region.

**Table 2.** Resistance and diffusion coefficients of the lithium ions of the two samples.

| Sample | $R_{ct}$ ($\Omega$) | $R_s$ | $D(Li^+)$ ($cm^2\ s^{-1}$) |
|---|---|---|---|
| LFP$-$a | 70.8 | 2.8 | $2.98 \times 10^{-14}$ |
| LFP$-$b | 56.7 | 1.3 | $3.40 \times 10^{-14}$ |

## 4. Conclusions

In summary, precursor $FePO_4$ with different grain sizes was prepared via the air oxidation precipitation method, and the electrochemical performance of the prepared $LiFePO_4$ was measured. The following conclusions can be drawn:

1.  The precursor $FePO_4$ prepared via the air oxidation precipitation method has high purity, the particle size is about 1 um, and the ratio of iron to phosphorus is close to 1.
2.  The reduction of the primary particle size to 92.7 nm significantly improves the rate performance of $LiFePO_4$. The discharge specific capacity of small grain LFP$-$b is about 155 mAh g$^{-1}$ at low current density. When the current density is increased to 10 C, the discharge specific capacity remains approximately 130 mAh g$^{-1}$ and the capacity retention is 83.5%. When the current density is restored to 0.2 C, the discharge specific capacity attenuation is 5.13%.
3.  Further studies show that grain refinement shortens the migration path of electrons and $Li^+$, which reduces the charge transfer resistance and promotes the diffusion of $Li^+$.

In the future, further research could be carried out to improve the oxidation efficiency of air. Alternatively, consideration should be given to using other cheaper raw materials for air oxidation to continue to reduce costs. Thus, this work provides a reference for reducing the production cost of $FePO_4$ and improving the electrochemical performance of $LiFePO_4$.

**Author Contributions:** Conceptualization, X.S.; Data curation, X.S.; Formal analysis, X.R.; Funding acquisition, Y.L. and Z.H.; Investigation, Z.Q.; Methodology, X.S.; Project administration, F.W.; Resources, Y.L.; Software, P.H.; Supervision, Y.C.; Validation, X.S.; Visualization, X.R.; Writing—original draft, X.S.; Writing—review & editing, Z.H. All authors have read and agreed to the published version of the manuscript.

**Funding:** This work was supported by the financial support from the Central South University Innovation-Driven Research Programme (Grant No. 2023CXQD053), Natural Science Foundation of Hunan Province China (Grant No. 2021JJ30823), National Natural Science Foundation of China (Grant No. 52274310). Government of Chongzuo, Guangxi Zhuang Autonomous Region (Fund No. FA2020011; FA20210713).

**Institutional Review Board Statement:** Not applicable.

**Informed Consent Statement:** Not applicable.

**Data Availability Statement:** Not applicable.

**Conflicts of Interest:** The authors declare no conflict of interest.

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
