# Peer review of "Refined Grain Enhancing Lithium-Ion Diffusion of LiFePO4 via Air Oxidation"

_coatings, doi:10.3390/coatings13061038_

Round 1

Reviewer 1 Report

This study used the air oxidation precipitation method to create FePO4 with various grain sizes, saving money on the cost of the oxidant. Additionally, the product is quite pure, and the ratio of  iron to phosphorus is very close to 1. Additionally, the impact of grain size on LiFePO4 was also explored. LiFePO4 made from small-grain FePO4 performs better in terms of rate.

This paper is interesting and covers an important topic. In overall, it is well written. Some of the comments are:

1) The content of Figure 1 cannot be read. It should be enhanced.

2) Same for Figure 3, 6 and 8.

3) Motivation of the study may be better highlighted in the Introduction.

4) Results and Discussions should be extended.

5) Some symbols and abbreviations are missing in the Nomenclature.

6) Conclusion can be expressed in the form of bullet points.

7) Some spelling errors should be avoided.

8) How the work can be extended in the future?

It is OK. Minor editing of English language required

Author Response

Dear Editor:

We would like to thank you and reviewers #1 and #2 for the opportunity and valuable comments you provided to improve this paper. We have addressed all the comments made by the reviewers point by point and have highlighted each change in red in the revised manuscript.

We wish to thank the editor and reviewers again for the considered comments and help and hope this revision could meet the requirements for publication in the journal.

With regards,

Yun Jiao Li (Email: [email protected]; [email protected])

Responses to Reviewers' Comments:

Reviewer #1:

Comments 1): The content of Figure 1 cannot be read. It should be enhanced.

Comments 2): Same for Figure 3, 6 and 8.

Response:

We sincerely thank the reviewers for their comment. We took a closer look at Figures 1, 3, 6 and 8 and found that the clarity of the images was low and the markings were indeed blurred. Therefore, we enhanced the clarity of the images and increased the markings of the annotations so that the information could be read.

Comments 3): Motivation of the study may be better highlighted in the Introduction.

Response:

Thanks to the reviewers for their valuable comments. Reading through the introduction section, it became apparent that the motivation for the research was not clearly highlighted. Therefore, the last paragraph of the introduction begins by mentioning the disadvantages of the existing iron phosphate preparation process, which consumes a large amount of oxidant, and the fact that there is less research combining cost reduction with improved electrochemical performance of lithium iron phosphate. These two reasons lead to the study of this paper, highlighting the motivation for the research. Modify as follows:

“However, industry usually uses ferrous sulphate as a raw material to prepare ferric phosphate, but the existing process must consume large amounts of oxidants such as hydrogen peroxide and sodium hypochlorite, which increases production costs to some extent. Moreover, few studies have combined reducing the production cost of FePO4 with improving the electrochemical performance of LiFePO4.”

Comments 4): Results and Discussions should be extended.

Response:

We are very grateful to the reviewer for critical comment. To solve this problem, we have extended some of the sections, mainly Figure 2,4,7. The expanded statements are located in lines 176-179, 197-199, 256-257 and 259-264 of the revised version.

Comments 5): Some symbols and abbreviations are missing in the Nomenclature.

Response:

Thank reviewer very much for challenging and valuable comment. We double-checked the symbols and abbreviations and made corrections where we found individual errors. As for the abbreviations, the main focus was on the names of journals in the references section. The CAS Source Index (CASSI) Search Tool was used to check the abbreviations of most of the journal titles. Only in the first case the abbreviation of the journal name was not found and the full name was used, so we asked for your help in correcting it accordingly.Modify as follows:

In the revised draft: “D(Li+)” in equation 1 is changed to “D”; “PDF#83-2092” in line 187 is changed to “PDF#83-2093”; “FE-SEM” in line 203 is changed to “SEM”.

Comments 6): Conclusion can be expressed in the form of bullet points.

Response:

Thank reviewer very much for challenging and valuable comment. The conclusion was expressed in the form of three main points. Modify as follows:

“In summary, precursor FePO4 with different grain sizes was prepared by air oxidation precipitation method, and the electrochemical performance of the prepared LiFePO4 was measured. The following conclusions can be drawn:

(1)The precursor FePO4 prepared by air oxidation precipitation method has high purity, the particle size is about 1um, and the ratio of iron to phosphorus is close to 1.

(2)The reduction of the primary particle size to 92.7 nm significantly improves the rate performance of LiFePO4. The discharge specific capacity of small grain LFP-b is about 155 mAhg-1 at low current density. When the current density is increased to 10 C, the discharge specific capacity still remains approximately 130 mAhg-1 and the capacity retention is 83.5%. When the current density is restored to 0.2 C, the discharge specific capacity attenuation is 5.13%.

(3)Further studies show that grain refinement shortens the migration path of electrons and Li+, which reduces the charge transfer resistance and promotes the diffusion of Li+.

In the future, further research could be carried out to improve the oxidation efficiency of air. Alternatively, consideration could be given to using other cheaper raw materials for air oxidation to continue to reduce costs. In a word, this work provides a reference for reducing the production cost of FePO4 and improving the electrochemical performance of LiFePO4”.

Comments 7): Some spelling errors should be avoided.

Response:

Thank the reviewer very much for careful comment. We went over the spelling of the words and corrected the misspellings. Modify as follows: In the revised version: “Lameller” to “Lamllar” in line 335.

Comments 8): How the work can be extended in the future?

Response:

Thanks to the reviewers for their valuable comments. Future work could further investigate the issue of how to improve the efficiency of air oxidation. Alternatively, combinations with other methods could be investigated to further reduce production costs. Modify as follows:

“In the future, further research could be carried out to improve the oxidation efficiency of air. Alternatively, consideration could be given to using other cheaper raw materials for air oxidation to continue to reduce costs”.

Reviewer 2 Report

The article is devoted to improvements in LiFePO4 technology for use as cathode material in batteries. Although it is stated what the improvements bring, it is not known what savings result from it. This requires supplementation. Nevertheless, the article contains interesting research material and is worth publishing.

As for the form of publication:

- markings on the axes of all drawings are not legible,

- for the sake of clarity, it is proposed to explain the notations FP-a, FP-b, LFP-a and LFP-b right at the beginning of chapter 2 and not in the text. This will make it easier to understand

- abbreviation HR-TEM (line 195) requires explanation

- formula (2) what does the symbol n mean?

-Table 2- what does Rs mean?

Author Response

Dear Editor:

We would like to thank you and reviewers #1 and #2 for the opportunity and valuable comments you provided to improve this paper. We have addressed all the comments made by the reviewers point by point and have highlighted each change in red in the revised manuscript.

We wish to thank the editor and reviewers again for the considered comments and help and hope this revision could meet the requirements for publication in the journal.

With regards,

Yun Jiao Li (Email: [email protected]; [email protected])

Responses to Reviewers' Comments:

Reviewer: 2

Reviewer(s)' Comments to Author:

Comments 1): The article is devoted to improvements in LiFePO4 technology for use as cathode material in batteries. Although it is stated what the improvements bring, it is not known what savings result from it. This requires supplementation. Nevertheless, the article contains interesting research material and is worth publishing.

Response:

We appreciate the reviewer’s important comments and help in improving the quality of this paper. In the revised version: lines 100-102 add what savings the method brings. Modify as follows: “This method not only saves on the cost of purchasing oxidants, but also reduces the cost of subsequent waste liquid disposal”.

Comments 2): markings on the axes of all drawings are not legible.

Response:

We are very grateful to the reviewer for valuable comment. We have enhanced the clarity of the image markers. The pictures in the revised version have been revised

Comments 3): for the sake of clarity, it is proposed to explain the notations FP-a, FP-b, LFP-a and LFP-b right at the beginning of chapter 2 and not in the text. This will make it easier to understand.

Response:

We are very grateful to the reviewer for valuable comment. We explain the meaning of FP-a, FP-b, LFP-a and LFP-b in Section 2. The location of the changes is in lines 121-122 and 129-130 of the revised version.

Comments 4): abbreviation HR-TEM (line 195) requires explanation

Response:

Thank the reviewer very much for careful comment. HR-TEM is short for high resolution TEM, namely, high resolution transmission microscope. Explained in lines 205-206 of the revised draft.

Comments 5): formula (2) what does the symbol n mean?

Response:

Thank the reviewer very much for careful comment. We forgot to clarify what n stands for. The n is the number of electrons gained or lost in the reaction, in this n=1. Added in line 284 of the revised version.

Comments 6): Table 2- what does Rs mean?

Response:

Thank the reviewer very much for careful comment. We forgot to clarify what Rs and Rct stands for. The Rs represents electrolyte impedance and Rct represents interface charge transfer impedance.Added in the revised version in lines 277-278.

Round 2

Reviewer 1 Report

It can be accepted in this form.

Author Response

I am very honoured that the reviewers were satisfied with my revisions. I would like to express my deepest gratitude to you.
